# Examining risk factors for self-harm and suicide in LGBTQ+ young people: a systematic review protocol

A Jess Williams  ,[1,2] Jon Arcelus,[3] Ellen Townsend,[2] Maria Michail[1]

[1]Institute for Mental Health, University of Birmingham, Birmingham, UK
[2]Self-Harm Research Group, University of Nottingham, Nottingham, UK
[3]Institute of Mental Health, University of Nottingham, Nottingham, UK

**Correspondence to**
A Jess Williams;
a.williams.10@pgr.bham.ac.uk

## ABSTRACT

**Introduction** Young people who identify as Lesbian, Gay, Bisexual, Transgender, Queer or Questioning (LGBTQ+) are at increased risk for self-harm, suicide ideation and behaviours. However, there has yet to be a comprehensive understanding of what risk factors influence these behaviours within LGBTQ+ young people as a whole. The purpose of this systematic review is to examine risk factors associated with self-harm, suicidal ideation and behaviour in LGBTQ+) young people.

**Methods and analysis** A systematic review will be conducted, conforming to the reporting guidelines of the Preferred Reporting Items for Systematic Reviews and Meta-Analyses statement recommendations. Electronic databases (MEDLINE, Scopus, EMBASE, PsycINFO and Web of Science) will be systematically searched for cross-sectional, prospective, longitudinal, cohort and case–control designs which examine risk factors for self-harm and/or suicidal ideation and behaviour in LGBTQ+ young people (aged 12–25 years). Only studies published in English will be included. No date restrictions will be applied. Study quality assessment will be conducted using the original and modified Newcastle-Ottawa Scales. Meta-analysis or narrative synthesis will be used, dependent on findings.

**Ethics and dissemination** This is a systematic review of published literature and thereby ethical approval was not sought. The review will be submitted to a peer-reviewed journal, be publicly disseminated at conferences focusing on mental health, self-harm and suicide prevention. The findings will also be shared through public engagement and involvement, particularly those related to young LGBTQ+ individuals.

**PROSPERO registration number** CRD42019130037.

## Strengths and limitations of this study

► This is the first systematic review of risk factors for both self-harm as an individual behaviour, and suicidal ideation and suicidal behaviour among Lesbian, Gay, Bisexual, Transgender, Queer or Questioning young people.

► The protocol has been written following the Preferred Reporting Items for Systematic Reviews and Meta-Analyses Protocol guidelines (2015).

► The robust search strategy was conducted with input from an academic skills specialist to ensure replication across all databases, and capture a large range of papers.

► The systematic review will follow the Cochrane Review guidelines.

► A limitation of this systematic review is the exclusion of studies which are not published in English; this may bias findings against non-English speaking countries.

## INTRODUCTION

Suicide, intentionally ending one's own life,[1] is one of the leading causes of death within young people.[2 3] Globally, suicide accounts for around 131 441 deaths between the ages of 15 and 24.[2] Rates of self-harm, the self-injury or poisoning of one's self irrespective of suicidal intent,[4] are on the rise among young people.[5] Data from the 2014 UK Adult Morbidity Survey state that 25.7% of women and 9.7% of men were likely to have self-harmed between the ages of 16 and 24 years.[6] Approximately 50% of those who die by suicide have previously self-harmed, making self-harm one of the strongest predictors of suicide.[7–9]

Among people who identify under the umbrella term LGBTQ+ (Lesbian, Gay, Bisexual, Transgender, Queer or Questioning), this being how one self-identifies as their sex, sexuality, gender identity and gender expression,[10] there are elevated rates of suicidal ideation (thoughts of self-harm, the consideration or planning of suicide[1]) and suicidal behaviour.[11–13] King et al's 2008 systematic review found that lesbian, gay and bisexual (LGB) individuals were found to be at higher risk for both suicidal behaviour and ideation compared with heterosexuals.[11] Internationally, this has been estimated to be between four and eight times higher in LGB and transgender young people when compared with the heterosexual and cisgender peers.[14] While as a broad group, sexual and gender minorities have high levels of suicidal ideation and behaviour, among transgender people (individuals who do not present or identify with their sex assigned at

birth[15]) these rates are even greater. Indeed, lifetime risk for suicide attempt is estimated to be between 22% and 43%, with 9%–10% having made an attempt in the past 12 months.[16–20]

Previous research has examined potential risk factors for self-harm and suicide in young people in general populations. These risk factors range from demographic characteristics such as being female, being a younger adult or adolescent, having fewer qualifications,[21] to childhood abuse and neglect.[22 23] Additional risks such as bullying and academic pressure have been linked to suicidal behaviour in young people,[9] as well as any mental health disorder.[24] Other factors which relate to the risk of self-harm repetition include troubled relationships with family members, social isolation, poor academic performance, alcohol and drug misuse, and depression.[25]

Given the heightened risk of suicide attempt within LGBTQ+,[11–13 26] it would be pertinent to explore which factors specifically pose as a risk to this population. Although some of these risk factors may overlap with those from non-LGBTQ+ populations (eg, depression, substance or alcohol misuse),[11 27] there are also unique risk factors to sexual and gender minorities. For example, Clements-Nolle et al[27] stated that experiences of abuse, discrimination or harassment due to an individual's gender identity or presentation were also linked to high levels of attempted suicide. Meyer[28 29] suggested that additional risk within homosexual communities may be related to the high levels of stigma, prejudice and discrimination which thereby impacts individual's mental well-being. Victimisation or discrimination due to sexual orientation and gender identity is particularly common within LGBTQ+ populations.[30–33] In addition, among LGBTQ+ young people, these experiences were found to be significantly associated with suicidal ideation.[34 35] It is suggested that young people may internalise these experiences of public stigma in relation to being LGBTQ+, known as internalised homophobia,[36] which can lead to adverse impacts on their self-perception and beliefs,

which could then enhance suicidal ideation, behaviour or self-harm.[37]

## Why this review is important

The goal of this systematic review is to comprehensively examine risk factors for self-harm and suicide ideation and behaviour within LGBTQ+ young people. Identifying and understanding how these factors relate to self-harm and suicide could allow future research to address specific risks and streamline potential studies to target the mental health needs of this population. Previous reviews regarding various dimensions of self-harm[22 23 38–40] have focused on a specific subgroup, such as the prevalence within transgender population[41] or sexual orientation.[42] Whereas, few have considered both self-harm and suicide in relation to LGBTQ+ populations as a broad group.[11 42–44] Thus, this review not only takes a dimensional approach to include self-harmful thoughts and behaviour with and without intent of suicide, thereby covering a wide spectrum of suicide and self-harm, but also includes all sexual and gender minorities.

## Objectives

1. To examine risk factors associated with self-harm and suicide ideation and behaviour in LGBTQ+ young people.
2. To examine whether there is a difference between sexual orientation minority young people and gender identity minority young people in the type of risk factors for self-harm, suicidal ideation and behaviour.

## METHODS

A systematic review of empirical quantitative studies which examine risk factors for self-harm and suicidal ideation and behaviour in LGBTQ+ young people will be conducted. The search considered all studies up to 1 April 2019, a summary of eligibility criteria is shown in table 1. This protocol follows the Preferred Reporting Items for Systematic Reviews and Meta-Analyses Protocol

**Table 1** Inclusion and exclusion criteria for papers

| Inclusion criteria | Exclusion criteria |
|---|---|
| ► Peer reviewed studies.<br>► Any geographical location.<br>► English language.<br>► Empirical quantitative studies, following cross-sectional, prospective, longitudinal, cohort and case–control designs.<br>► Studies must consider factors associated with or predictive of self-harm, suicidal ideation or suicidal behaviour.<br>► Participants must be young people (12–25 years).<br>► Participants who are identified or self-identified as any sexual or gender minority or member of LGBTQ+.<br>► Participants who have had an experience of self-harm, suicidal ideation or behaviour. | ► Non-peer reviewed literature.<br>► Not English language.<br>► Grey literature such as theses, dissertation or conference proceedings.<br>► Articles such as commentaries, reviews, editorial or opinion pieces.<br>► Empirical qualitative studies.<br>► Sample not aged between 12 and 25 years, for example, adults 26 years and above or children 12 years and under.<br>► Participants who are identified as heterosexual or not part of sexual or gender minority.<br>► Participants who have no experience of self-harm, suicidal ideation or suicidal behaviour. |

LGBTQ+, Lesbian, Gay, Bisexual, Transgender, Queer or Questioning.

guidelines,[45] presented in the online supplementary file 1. Study quality will be assessed by the Newcastle-Ottawa Scale (NOS)[46] and the adapted version for cross-sectional studies.[47]

### Eligibility criteria

See table 1 for summary:

### Types of studies

These include empirical quantitative peer-reviewed studies following cross-sectional, prospective, longitudinal, cohort and case–control designs which examine risk factors for self-harm and/or suicidal ideation and behaviour in LGBTQ+ young people. Risk factors being identified as significant predictors, mediators or moderators which influence a self-harm or suicidal outcome. These papers may specifically focus on one subgroup of the LGBTQ+ umbrella (eg, transgender) or look across groups (eg, LGB). Mixed-method papers which have applicable extractable information will also be included. Grey literature such as theses, dissertations or conference proceedings will not be included. Commentary, reviews, editorial or opinion pieces will also be excluded. All included studies must be available as full-text and peer-reviewed in the English language.

### Types of participants

Participants between the ages of 12 and 25 years old, who identify as LGBTQ+ referring to how one self-identifies as their sex, sexuality, gender identity and gender expression.[10] This age range was selected to mirror papers within the field that consider young people up to the age of 25 years,[9 48] with the lower limit extended to 12 years to include the adolescent period. Participants will also have had experiences of self-harm (self-injury or poisoning irrespective of suicidal intention[4]), suicide ideation (this can include thoughts of self-harm, the consideration or planning of a suicidal attempt, eg method[1]) or suicidal behaviour (intentionally trying to end one's own life[1]). Alternatively, if participants met the above inclusion criteria as a subgroup, this data will be extracted.

### Types of outcome measures

Studies will be included if they used a measure for self-harm, suicidal ideation or suicidal behaviour. The measurements of self-harm (any, with intent and without intent) and suicidal behaviour; attempt or death, will be taken as binary variables; present or absence. Continuous scales for suicidal ideation will be converted to binary variables by considering the original scale threshold. This will act as a cut-off point, if the population mean for ideation is above this threshold, ideation will be considered present, whereas below the threshold, ideation will be considered absent. The psychometric measures of these outcomes will be extracted for descriptive purposes. This can be specific tools for considering suicide, for example, Beck Scale for Suicide Ideation;[49] Motto's Risk Estimator for Suicide;[50] Positive and Negative Suicide Ideation;[51] Columbia-Suicide Severity Rating Scale;[52] or

self-harm, for example, Self-Harm Inventory;[53] Inventory of Statements about Self-injury;[38] Self-Injurious Thoughts and Behaviour Interview Short Form.[54] Also included will be self-harm or suicide items from general scales such as items 102–105 from the Mental Health History Form (Boudewyn and Liem, 1995; Mental Health History Form) or individual questions regarding self-harm or suicide had ever been considered or acted on[55] and clinician reports.

### Search strategy

The search was limited to English language and was run up to 1 April 2019. No date restrictions were applied. The following electronic bibliographic databases were searched: MEDLINE, Scopus, EMBASE, PsycINFO and Web of Science for peer-reviewed publications which examine risk factors for self-harm or suicide within LGBTQ+ young people. The search strategy can be found in the online supplementary file 2. This was developed in collaboration with an academic skill specialist from the University of Birmingham library to ensure the robustness of the search. Reference lists of eligible papers and conduct citation searches of key papers were explored to identify additional reports. Before the final draft of the systematic review is completed, a second search will be conducted to allow for any additional studies to be identified.

### Study records

#### Selection process

The search strategy will retrieve titles and/or abstracts which will be screened by two independent researchers to identify studies which potentially meet the inclusion criteria outlined above. The full-texts of the studies will be retrieved and independently assessed for eligibility. Any disagreement or uncertainty over the eligibility of particular studies will be resolved through discussion by two independent researchers.

#### Data management and collection

Rayyan QCRI (https://rayyan.qcri.org/welcome), the online systematic review tool, will be used to manage and screen all retrieved papers. AJW will be the solo team member responsible for adding or amending paper records in Rayyan, as well as identifying and removing duplicates. A second independent reviewer will be given access to all titles and abstracts, within Rayyan. They will be able to make their own decision as to whether to include or exclude a paper, blind to AJW's decisions. AJW will also be blind to the decisions of the second reviewer until all titles and abstracts are reviewed. Rayyan will provide information on the original number of titles screened, duplications, those excluded at this stage and those included titles at this stage.

Those papers which reach full-text screening will be managed within Zotero (https://www.zotero.org/), the bibliographic software. These will be sectioned by those included at full-text screening, and final inclusion.

Reason for exclusion will be reported for each paper. AJW will be responsible for liaising with interlibrary loans and obtaining the full-text papers. Full-text screening will again be conducted by two researchers, blind to each other's decisions, who will resolve disagreements through discussion. If an agreement cannot be reached regarding a paper, this will be rated by a third researcher.

A prepiloted standardised data extraction tool will be adapted and used on included studies.[56] Extracted information will include (1) author and publication date; (2) study design and setting; (3) characteristics of participants (age, gender identity), and the studies' inclusion and exclusion criteria; (4) method of harm (self-harm, suicidal behaviour or ideation); (5) factors associated with or predictive of self-harm or suicidal ideation and behaviour (including clinician diagnosis, subscales or validated scales used to assess these items) and (6) information relating to risk of bias for the individual study.

### Risk of bias (quality assessment)

The overall quality of each study will be evaluated by two independent researchers using the NOS[46] and the NOS adapted for cross-sectional studies.[47] Alternative study designs, which are not covered in these versions of NOS, will be assessed using a prepiloted version developed for a previous systematic review.[56] The level of agreement will be demonstrated by Cohen's Kappa and a Prevalence and Bias-Adjusted Kappa score.

### Data synthesis and assessment of heterogeneity

Included studies will be presented by grouping risk types together (eg, demographic, psychosocial, psychiatric or mental health). The descriptive characteristics of each will be presented in tabular format, and the risks and outcomes of which will be discussed using narrative synthesis, following the Economic and Social Research Council guidelines.[57] The analysis will take place over four steps; (1) developing a theory of which factors influence LGBTQ+ young people, (2) synthesising the findings, (3) exploring whether there are relationships within the data and what these might be and (4) assessing the robustness of the synthesis.

If possible, a meta-analysis of risk factors for self-harm and suicidal ideation and behaviour using RevMan 5 (https://community.cochrane.org/help/tools-and-software/revman-5/revman-5-download) may be conducted by pooling data on risk factors using a random-effects model with the assumption that populations have various effect sizes and weights studies from in-study and between-study variance. These risks will be pooled based on the analysis which has taken place in the original paper, if a risk appears in more than one type of analysis, it will be included in both analyses. These will consider the extent of heterogeneity, which will be determined using the $I^2$ statistic. From this, a summary effect will be presented, associated 95% CI and p values. OR with 95% CIs will present the overall synthesised measure of effect size.

Publication bias will be assessed using funnel plots and Harbord-Egger test.[58]

Additionally, consideration will be given to subgroup analysis if data allows, exploring heterogeneity by sexual and gender minorities; such that these studies are split by LGBQ and transgender; as even within this population it is recognised that the transgender population have high levels of suicide risk and self-harm.[41 48] This will allow us to determine which risk factors are similar within sexual and gender minorities, and those that unique to the transgender population. Further examination of risk factors will be conducted by age difference, split by 12–18 years and 19–25 years.

The quality of the included studies will be considered when synthesising and analysing the findings in terms of the strength of evidence which they provide. A sensitivity analysis will be run to assess the robustness of included studies,[59] following the removal of papers which score lower than 70% on quality assessment scales.

### Patient and Public Involvement

No patients involved.

## ETHICS AND DISSEMINATION

It will be submitted to a peer-reviewed journal, be publicly disseminated at conferences focusing on mental health, self-harm and suicide prevention. Findings will also be shared through public engagement and involvement, particularly considering young LGBTQ+ individuals.

## DISCUSSION

This systematic review will be the first to provide a rich, holistic account of the existing evidence of risk factors for self-harm and suicidality within a broad sample of LGBTQ+ young people. The synthesis of these findings will assess the prevalence of particular factors which impact this population, which may not be relevant to non-sexual or gender minority young people. However, when pooling studies within a meta-analysis, it is possible that the variation between papers may cause a challenge for this synthesis. This may be related to the broad approach in which we are approaching the systematic review, nonetheless we feel that this approach offers many values. Such as, exploring the similarities and differences of risk factors by subgroup analysis, via sexual orientation and gender identity or age, could be used as a valuable stepping stone when considering LGBTQ+ research and promoting more applicable research aims.

The quality and strength of the evidence will be rigorously assessed, which could be used to inform future research targeting these particular risks. This could ultimately help to inform self-harm and suicidality prevention within LGBTQ+ young people. It is anticipated that the findings of this review will be of interest to the wider academic and clinical community, policy-makers, young

people who identify as a sexual or gender minority, those with experience of self-harm or suicide.

**Contributors** AJW, JA, ET and MM conceptualised the study. AJW developed the search strategy and conducted the literature search. AJW and MM wrote the first draft of the manuscript. AJW, JA, ET and MM reviewed, edited and approved the final manuscript.

**Funding** This work was supported by the Midlands Graduate School Economic and Social Research Council Doctoral Training Partnership Joint Studentship awarded to A Jess Williams.

**Competing interests** None declared.

**Patient consent for publication** Not required.

**Ethics approval** This is a systematic review of published literature and thereby ethical approval was not sought.

**Provenance and peer review** Not commissioned; externally peer reviewed.

**ORCID iD**
A Jess Williams http://orcid.org/0000-0002-3987-3824

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
