## [Reviewer comments · BMJ Open]

ARTICLE DETAILS

TITLE (PROVISIONAL)	A systematic review protocol examining risk factors for self-harm and suicide in LGBTQ+ young people
AUTHORS	Williams, A. Jess; Arcelus, J; Townsend, Ellen; Michail, Maria

VERSION 1 – REVIEW

REVIEWER	Jose A. Piqueras University Miguel Hernández of Elche, Spain
REVIEW RETURNED	07-Jun-2019

GENERAL COMMENTS	The authors assert in “STRENGTHS AND LIMITATIONS OF THIS STUDY”: “This is the first systematic review of risk factors for self-harm, suicidal ideation and behaviour among LGBTQ+ young people”. This is not truth. In 2017 there is a systematic review and meta-analysis focused on sexual orientation and suicidal behavior”: Miranda-Mendizábal, A., Castellví, P., Parés-Badell, O., Almenara, J., Alonso, I., Blasco, M. J., ... & Piqueras, J. A. (2017). Sexual orientation and suicidal behaviour in adolescents and young adults: systematic review and meta-analysis. The British Journal of Psychiatry, 211(2), 77-87. At least the authors should mention it, and give the rationale why this new systematic review is needed. At least I know the following related publications, the authors should consider: Blasco, Maria Jesús; Castellví, Pere; Almenara, José; Lagares, Carolina; Roca, Miquel; Sesé, Albert; Piqueras, José Antonio; Soto-Sanz, Victoria; Rodríguez-Marín, Jesús; Echeburúa, Enrique; Predictive models for suicidal thoughts and behaviors among Spanish University students: rationale and methods of the UNIVERSAL (University & mental health) project, BMC Psychiatry, 16,1,1,2016, BioMed Central Castellví, P., Miranda-Mendizábal, A., Parés-Badell, O., Almenara, J., Alonso, I., Blasco, M. J., ... & Piqueras, J. A. (2017). Exposure to violence, a risk for suicide in youths and young adults. A meta-analysis of longitudinal studies. Acta psychiatrica scandinavica, 135(3), 195-211. Castellví, P., Lucas-Romero, E., Miranda-Mendizábal, A., Parés-Badell, O., Almenara, J., Alonso, I., ... & Lagares, C. (2017).
---

	Longitudinal association between self-injurious thoughts and behaviors and suicidal behavior in adolescents and young adults: A systematic review with meta-analysis. Journal of affective disorders, 215, 37-48. Gili, M., Castellví, P., Vives, M., de la Torre-Luque, A., Almenara, J., Blasco, M. J., ... & Parés-Badell, O. (2018). Mental disorders as risk factors for suicidal behavior in young people: A meta-analysis and systematic review of longitudinal studies. Journal of affective disorders Miranda-Mendizabal, A., Castellví, P., Parés-Badell, O., Alayo, I., Almenara, J., Alonso, I., ... & Lagares, C. (2019). Gender differences in suicidal behavior in adolescents and young adults: systematic review and meta-analysis of longitudinal studies. International journal of public health, 64(2), 265-283. Miranda-Mendizabal, A., Castellví, P., Parés-Badell, O., Almenara, J., Alonso, I., Blasco, M. J., ... & Piqueras, J. A. (2017). Sexual orientation and suicidal behaviour in adolescents and young adults: systematic review and meta-analysis. The British Journal of Psychiatry, 211(2), 77-87. Soto-Sanz, V., Castellví, P., Piqueras, J. A., Rodríguez-Marín, J., Rodríguez-Jiménez, T., Miranda-Mendizabal, A., ... & Cebria, A. (2019). Internalizing and externalizing symptoms and suicidal behaviour in young people: a systematic review and meta-analysis of longitudinal studies. Acta psychiatrica Scandinavica.
--	--

REVIEWER	John McAloon University of Technology Sydney Australia
REVIEW RETURNED	04-Jul-2019

GENERAL COMMENTS	Abstract The methods section of the abstract, together with Para commencing Line 194., suggests that “A systematic review and meta-analysis will be conducted”, “a meta-analysis will be performed if appropriate” and “will be conducted”. It might be an idea, and certainly within the limits of PROSPERO registration, to run preliminary searches to ascertain whether you are in a position to conduct a meta-analysis. Good to see PROSPERO registration Introduction One of the most important themes alluded to here first in Para 4 (lines 83-96) is whether qualitatively distinct factors of risk for self-harm or suicide can be identified that are unique to LGBTQI people. For instance, were we to identify chronic psychotic illness as one such factor in LGBTQI people, we would need to be able to demonstrate that it is qualitatively different from the chronic psychotic illness experienced by non LGBTQI people as a result of their LGBTQI status or prefaces. I think it important that the introduction demonstrate this. Yes, Meyer and colleagues suggest that there is added risk, but is it qualitatively different risk from that experienced by other people who experience abuse, discrimination, harassment, stigma and prejudice? Line 94. Reference to internalised homophobia might be appropriate here, for instance Puckett, J. A., Newcomb, M. E., Ryan, D. T., Swann, G., Garofalo, R., & Mustanski, B. (2017). Internalized homophobia and
---

	perceived stigma: A validation study of stigma measures in a sample of young men who have sex with men. Sexuality Research and Social Policy, 14(1), Lines 106-107 There is comment in existing research that the range of variability across functions, motives and associations for self-harm and suicide, is large. I wonder if the “dimensional” approach to this subject needs further justification. On the face of it, it seems a little too broad and may, as a result, appear contrary to current evidence. Finding a systematic review with a single aim is rare these days, it is more common to find a primary aim and one or more secondary aims. The review might consider taking the opportunity to extend itself beyond a single aim. For instance, is there an opportunity to look into the functions of self-harm/suicide in LBGTQI people as distinct from those in other populations? Methods The idea of factors of risk may warrant additional attention. For instance, a risk factor might be refined in definition as a mediator, a moderator, a predictor. Again, for the sake of replicability, this term may benefit from some definitional parameters. Line 132, 133, 134 need to close the brackets? Line 136. Does the nature of the measure matter? For instance, you might specify “psychometric assessment measures with previously published psychometric properties” otherwise we don’t know the parameters necessary for replication and thus are limited in our understanding of systematic Line 137. “Consideration of number...” is it appropriate to collapse this to a binary variable if there is good evidence that there is a strong positive association between number of self-harm events and completed suicide? Surely this is a factor risk albeit not unique but applicable to LBGTQI people. Line 157. The review should report the final search as past tense as it has already happened by the point at which the manuscript has been written. Line 163. Who is the discussion between? Study author or independent? Line 178. And 188. Its common to provide a metric of agreement between raters (Kappa, ICC etc) Line 198: “or” should be ‘of’ Line 201. Subgroyp analyses, if possible would be a valuable addition to the review. Discussion Very brief & OK References Good
--	---

REVIEWER	Sophie Epstein South London and Maudsley NHS Foundation Trust and King's College London, UK
REVIEW RETURNED	05-Jul-2019

GENERAL COMMENTS	Many thanks for sending this protocol for review. It seeks to answer an important question and will add important information to the evidence base on risk factors for self-harm and suicide in vulnerable groups of young people. I have some comments and suggestions as follows: Abstract: 1. Would it be useful to spell out LGBTQ+ in full the first time?
---

	2. In the methods, it would be helpful to have a definition of what the authors mean by 'young people' – ie an age range. Introduction: 3. The comment that self-harm is on the rise among young people needs a reference. 4. The adult morbidity survey is mentioned but it is not clear whether this is an international or national (which country?) study. 5. I am not sure about the definition of suicidal ideation. In many cases, it would not include thoughts of self-harm (unless that self-harm was with suicidal intent). I am not familiar with the DSM5 definition however if that is indeed the definition, it may need to be clarified that it is defined differently in different contexts. 6. Perhaps transgender could be spelled out in full rather than abbreviated to 'T' 7. The following paragraph states that risk factors for self-harm in young people include 'being younger'. This needs to be clarified as self-harm is less common in children under 12 than it is in adolescents. Methods 8. The authors state that they plan to use the Newcastle-Ottawa Scale. This scale is likely to need adapting to some extent to be relevant to the review it is being applied to. For example, it requires the most important covariates to be identified, the most appropriate methods of case ascertainment, what is considered a reasonable sample size and response rate etc. These will vary depending on the study. It would be helpful therefore, if the versions of the NOS to be used in this particularly study, with adaptations, were included as a supplementary file. 9. The age range stated is 12-15. Could the authors include a sentence to justify this choice of age range and also state whether studies which overlap this age range but also include older or younger participants will be included or excluded. 10. This paragraph also states that 'participants will also have had experiences of self-harm' – presumably this is not the case, as only some of the participants will go on to experience the outcome (self-harm) and others will not. 11. The paragraph on types of outcome measures needs clarification. Will all outcomes be converted to binary variables? (ie presence or absence of self-harm or suicidal behaviour). What will be done with the suicide scales which are continuous measures – used as continuous variables or converted to binary? 12. Search strategy – will the authors also  Ask experts to identify any additional papers Conduct forward citation searching of included papers Hand search any particular relevant journals? 13. Selection process – presumably here, as well as 'resolving through discussion' a third person would be consulted? (This is mentioned later but not here). 14. Data synthesis: This section needs clarity in view of the fact that the authors will be including studies which are heterogeneous in terms of study design and outcome measures, and also that multiple different risk factors will be considered.  As per point number 11, could the authors clarify what the effect measures will be. If all outcomes are binary, will this be odds ratios. If not binary then what will be the effect measure. At present, there is no description of how the authors will decide which studies can be pooled in meta-analysis (only those with the
--	--

	same study design, outcome measure, exposures?) and whether univariate or multivariable results be pooled. c. Will there be a separate meta-analysis for each risk factor, and how will the authors determine when risk factors can be 'combined' e.g. would alcohol abuse/substance misuse/illicit drug use be considered one risk factor or several? I don't think it is necessary to list all the potential risk factors and how they will be managed, but it would be useful to include something about how this issue will be approached. d. I am unclear what the authors mean by 'a sensitivity analysis will be run to assess the robustness of the studies'. Does this mean a sensitivity analysis where studies with lower quality ratings will be removed from the analysis? Supplementary file 2 (search strategies) 15. The Embase search strategy does not appear to include all the terms listed on the previous page. In fact, it includes almost exclusively thesaurus terms and not keywords and does not include the second set of terms listed on the previous page (pertaining to risk factors/mechanisms etc). If possible, I would suggest that these terms not be included (as it appears they have not been in the Embase search) as their inclusion may limit the sensitivity of the search. And the inclusion of keywords as well as thesaurus terms is important to ensure sensitivity of the search.
--	--

VERSION 1 – AUTHOR RESPONSE

Reviewer 1:

The authors assert in “STRENGTHS AND LIMITATIONS OF THIS STUDY”: “This is the first systematic review of risk factors for self-harm, suicidal ideation and behaviour among LGBTQ+ young people”.

This is not truth. In 2017 there is a systematic review and meta-analysis focused on sexual orientation and suicidal behavior”:

Miranda-Mendizábal, A., Castellví, P., Parés-Badell, O., Almenara, J., Alonso, I., Blasco, M. J., ... & Piqueras, J. A. (2017). Sexual orientation and suicidal behaviour in adolescents and young adults: systematic review and meta-analysis. *The British Journal of Psychiatry*, 211(2), 77-87.

At least the authors should mention it, and give the rationale why this new systematic review is needed.

Thank you for highlighting this paper, we have since included this in our protocol.

Line 108, page 4.

Although previous studies have considered sexual orientation, they have not included transgender young people within the same group. Our review aims to include all the LGBTQ+ population in that respect we believe this is the first study that includes the whole spectrum of sexual orientation and gender identity. We believe that the added value of considering both sexual orientations and gender identities allows for us to gain a further understanding of those risk factors which transcend across groups, and allow us to pinpoint where the differences may be.

At least I know the following related publications, the authors should consider:

Blasco, Maria Jesús; Castellví, Pere; Almenara, José; Lagares, Carolina; Roca,

Miquel; Sesé, Albert; Piqueras, José Antonio; Soto-Sanz, Victoria; Rodríguez-Marín, Jesús; Echeburúa, Enrique; Predictive models for suicidal thoughts and behaviors among Spanish University students: rationale and methods of the UNIVERSAL (University & mental health) project, *BMC Psychiatry*, 16,1,1,2016, BioMed Central

Castellví, P., Miranda-Mendizábal, A., Parés-Badell, O., Almenara, J., Alonso, I., Blasco, M. J., ... & Piqueras, J. A. (2017). Exposure to violence, a risk for suicide in youths and young adults. A meta-analysis of longitudinal studies. *Acta psychiatrica scandinavica*, 135(3), 195-211.

Castellví, P., Lucas-Romero, E., Miranda-Mendizábal, A., Parés-Badell, O., Almenara, J., Alonso, I., ... & Lagares, C. (2017). Longitudinal association between self-injurious thoughts and behaviors and suicidal behavior in adolescents and young adults: A systematic review with meta-analysis. *Journal of affective disorders*, 215, 37-48.

Gili, M., Castellví, P., Vives, M., de la Torre-Luque, A., Almenara, J., Blasco, M. J., ... & Parés-Badell, O. (2018). Mental disorders as risk factors for suicidal behavior in young people: A meta-analysis and systematic review of longitudinal studies. *Journal of affective disorders*

Miranda-Mendizabal, A., Castellví, P., Parés-Badell, O., Alayo, I., Almenara, J., Alonso, I., ... & Lagares, C. (2019). Gender differences in suicidal behavior in adolescents and young adults: systematic review and meta-analysis of longitudinal studies. *International journal of public health*, 64(2), 265-283.

Miranda-Mendizábal, A., Castellví, P., Parés-Badell, O., Almenara, J., Alonso, I., Blasco, M. J., ... & Piqueras, J. A. (2017). Sexual orientation and suicidal behaviour in adolescents and young adults: systematic review and meta-analysis. *The British Journal of Psychiatry*, 211(2), 77-87.

Soto-Sanz, V., Castellví, P., Piqueras, J. A., Rodríguez-Marín, J., Rodríguez-Jiménez, T., Miranda-Mendizábal, A., ... & Cebria, A. (2019). Internalizing and externalizing symptoms and suicidal behaviour in young people: a systematic review and meta-analysis of longitudinal studies. *Acta psychiatrica Scandinavica*.

Thank you for highlighting these papers. We have included those that add to the arguments within this manuscript.

Reviewer 2:

Abstract The methods section of the abstract, together with Para commencing Line 194., suggests that “A systematic review and meta-analysis will be conducted”, “a meta-analysis will be performed if appropriate” and “will be conducted”. It might be an idea, and certainly within the limits of PROSPERO registration, to run preliminary searches to ascertain whether you are in a position to conduct a meta-analysis.

We have aimed to make this clearer within the abstract (line 34, 41-42, page 2) and data synthesis and assessment of heterogeneity (line 206-7, page 7).

Preliminary searches were indeed run which suggested there was potential for a meta-analysis given the number of quantitative cross-sectional studies. However, paper reporting is variable and this cannot yet be stated with certainty how findings will need to be interpreted.

Introduction

One of the most important themes alluded to here first in Para 4 (lines 83-96) is whether qualitatively distinct factors of risk for self-harm or suicide can be identified that are unique to LGBTQI people. For instance, were we to identify chronic psychotic illness as one such factor in LGBTQI people, we would need to be able to demonstrate that it is qualitatively different from the chronic psychotic illness experienced by non LGBTQI people as a result of their LGBTQI status or prefaces. I think it important that the introduction demonstrate this. Yes, Meyer and colleagues suggest that there is added risk, but is it qualitatively different risk from that experienced by other people who experience abuse, discrimination, harassment, stigma and prejudice?

It is not our aim to compare LGBTQ risk factors with those of heterosexual cisgender individuals. It is simply to understand what risk factors are relevant to an LGBTQ population of young people.

The review may show that there are similar risk factors among different groups, it may also be that risk factors are not hugely different to those who also suffer from stigma, harassment, discrimination, but we will not be sure of this until the literature review is completed. The introduction of this review describes what is known about risk factors in young people in general and that LGBTQ+ young people have experiences, which may be important, stigma or discrimination, as risk factors. I hope this clarifies this point for the reviewer.

Line 94. Reference to internalised homophobia might be appropriate here, for instance Puckett, J. A., Newcomb, M. E., Ryan, D. T., Swann, G., Garofalo, R., & Mustanski, B. (2017). Internalized homophobia and perceived stigma: A validation study of stigma measures in a sample of young men who have sex with men. *Sexuality Research and Social Policy*, 14(1), Lines 106-107

This reference has been incorporated.

Line 99, page 3

There is comment in existing research that the range of variability across functions, motives and associations for self-harm and suicide, is large. I wonder if the “dimensional” approach to this subject needs further justification. On the face of it, it seems a little too broad and may, as a result, appear contrary to current evidence.

For this protocol and study, we use the National Institute for Health and Clinical Excellence definition of self-harm, which considers self-harm irrespective of suicidal intention (NICE, 2011). This allows for the dimensional approach considering self-harm, suicidal ideation and behaviour within this review, reinforced by Orlando et al., (2015) taxometric investigation which confirms the dimensional nature of self-harm.

Dependent on findings, there is potential to synthesize findings by self-harm, suicidal ideation, and suicidal behaviour separately, following the excellent example of Robinson, Hetrick, & Martin (2011). Robinson, J., Hetrick, S. E., & Martin, C. (2011). Preventing suicide in young people: systematic review. *Australian and New Zealand journal of psychiatry*, 45(1), 3-26.

Finding a systematic review with a single aim is rare these days, it is more common to find a primary aim and one or more secondary aims. The review might consider taking the opportunity to extend itself beyond a single aim. For instance, is there an opportunity to look into the functions of self-harm/suicide in LGBTQI people as distinct from those in other populations?

Thank you for this comment, we have included a secondary aim now which is to compare the different type of risk factors which may impact sexual orientation minority young people and gender identity minority young people.

Line 116-118, page 4.

Methods

The idea of factors of risk may warrant additional attention. For instance, a risk factor might be refined in definition as a mediator, a moderator, a predictor. Again, for the sake of replicability, this term may benefit from some definitional parameters.

Thank you for this comment, the definition is how we are considering risk factors and have therefore included this.

Line 131-133, page 4.

Line 132, 133, 134 need to close the brackets?

These have been added.

Line 142-144, page 5.

Line 136. Does the nature of the measure matter? For instance, you might specify “psychometric assessment measures with previously published psychometric properties” otherwise we don’t know the parameters necessary for replication and thus are limited in our understanding of systematic We have included the psychometric assessment for self-harm, suicidal ideation, and suicidal behaviour as these are variable between studies and often give more descriptive information about what the study was specifically targeting. We believe that this will make the systematic review more transparent and reliable.

Line 137. “Consideration of number...” is it appropriate to collapse this to a binary variable if there is good evidence that there is a strong positive association between number of self-harm events and completed suicide? Surely this is a factor risk albeit not unique but applicable to LGBTQI people.

We agree with this point and have adjusted the manuscript accordingly.

Line 147-148, page 5.

Line 157. The review should report the final search as past tense as it has already happened by the point at which the manuscript has been written.

This has been adjusted.

Line 122-123, page 4.

Line 160-161, page 5.

Line 163. Who is the discussion between? Study author or independent?

Discussions will be conducted between two independent researchers who will separately review paper titles, abstracts, full-text and then extraction separately.

Line 173-174, page 6.

Line 178. And 188. Its common to provide a metric of agreement between raters (Kappa, ICC etc)

Thank you for this comment. This will be conducted using a Cohen’s Kappa and a Prevalence and Bias-adjusted Kappa score. Cohen’s Kappa is more widely used and therefore may be easier for readers to interpret, however the PABAK offers more precise measurements of reviewer agreements.

Line 202-203, page 6.

Line 198: “or” should be ‘of’

This has been adjusted.

Line 201. Subgroup analyses, if possible would be a valuable addition to the review.

Thank you, we still aim to do this.

Page 7.

Reviewer 3:

Abstract:

1. Would it be useful to spell out LGBTQ+ in full the first time?

We have now included this.

Line 33, page 2.

2. In the methods, it would be helpful to have a definition of what the authors mean by 'young people' – ie an age range.

This has also been included.

Line 39, page 2.

Introduction:

3. The comment that self-harm is on the rise among young people needs a reference.

We have since added this.

Line 64, page 3.

4. The adult morbidity survey is mentioned but it is not clear whether this is an international or national (which country?) study.

This is a survey of the UK, we have added this for clarity.

Line 64, page 3.

5. I am not sure about the definition of suicidal ideation. In many cases, it would not include thoughts of self-harm (unless that self-harm was with suicidal intent). I am not familiar with the DSM5 definition however if that is indeed the definition, it may need to be clarified that it is defined differently in different contexts.

For the purposes of this study we have adopted the DSM5 definition, which considers self-harm as one of the potential options for suicidal ideation, along with thoughts of or planning an attempt. This is similar to the Centers for Disease Control and Prevention in the United States (2011) definition; "thoughts of engaging in suicide-related behaviour" which is discussed by Jobes & Joiner (2019). I have aimed to clarify the variation which may occur however.

Line 142, page 5.

6. Perhaps transgender could be spelled out in full rather than abbreviated to 'T'

This has been adjusted to reflect this comment.

Line 74, page 3.

7. The following paragraph states that risk factors for self-harm in young people include 'being younger'. This needs to be clarified as self-harm is less common in children under 12 than it is in adolescents.

I have clarified this as "younger adult or adolescent" now.

Line 82, page 3.

Methods

8. The authors state that they plan to use the Newcastle-Ottawa Scale. This scale is likely to need adapting to some extent to be relevant to the review it is being applied to. For example, it requires the most important covariates to be identified, the most appropriate methods of case ascertainment, what is considered a reasonable sample size and response rate etc. These will vary depending on the study. It would be helpful therefore, if the versions of the NOS to be used in this particularly study, with adaptations, were included as a supplementary file.

We are using the published versions of the NOS and an adapted version of the NOS, which has previously been used in a large systematic review currently in the final stages of review. This adaptation considers potential sources of selection bias and exposure ascertainment. It is however the intellectual property of the University of Bristol, I am happy to provide contact details for the author on request.

9. The age range stated is 12-15. Could the authors include a sentence to justify this choice of age range and also state whether studies which overlap this age range but also include older or younger participants will be included or excluded.

For clarity, we have used the age range 12-25 years, not 12-15 years. This is mirrored by papers within the field that consider young people up to the age of 25 years (Marchant et al., 2019; Arcelus et al., 2016; Rodway et al., 2016; Hawton et al., 1999). We extended the lower limit to 12 years to include the adolescent period.

Authors will be contacted if they have a participant population which overlaps with this age range.

This will be based on how they have presented their descriptive and analytic data such that authors who have an age grouping which overlaps with our age range could easily answer whether the risks were significant for participants in our age range.

10. This paragraph also states that 'participants will also have had experiences of self-harm' – presumably this is not the case, as only some of the participants will go on to experience the outcome (self-harm) and others will not.

The whole sentence of this reads "self-harm, suicidal ideation or suicidal behaviour", rather than expecting all participants to have the same experiences. Thereby we acknowledge that it is not the case all participants will experience the same outcomes.

11. The paragraph on types of outcome measures needs clarification. Will all outcomes be converted to binary variables? (ie presence or absence of self-harm or suicidal behaviour). What will be done with the suicide scales which are continuous measures – used as continuous variables or converted to binary?

The measurement of self-harm (any, with intent, without intent), suicidal ideation and suicidal behaviour; attempt or death will be converted into binary variables; present or absence. The psychometric measures of these outcomes will be extracted for descriptive purposes.

Line 147-148, page 5.

12. Search strategy – will the authors also

- a. Ask experts to identify any additional papers
- b. Conduct forward citation searching of included papers
- c. Hand search any particular relevant journals?

Our search strategy was rather extensive within the databases. Alongside this, we have considered the citation lists of relevant articles and identified further key papers.

Line 165-166, page 6.

13. Selection process – presumably here, as well as 'resolving through discussion' a third person would be consulted? (This is mentioned later but not here).

This has been amended.

Line 173-174, page 6.

14. Data synthesis: This section needs clarity in view of the fact that the authors will be including studies which are heterogeneous in terms of study design and outcome measures, and also that multiple different risk factors will be considered.

- a. As per point number 11, could the authors clarify what the effect measures will be. If all outcomes are binary, will this be odds ratios. If not binary then what will be the effect measure.

Thank you for pointing this out; we are using binary outcomes, therefore we will be using odds ratios. I have added this into the text.

Line 212-213, page 7.

- b. At present, there is no description of how the authors will decide which studies can be pooled in meta-analysis (only those with the same study design, outcome measure, exposures?) and whether univariate or multivariable results be pooled.

Our findings will be very data dependent, therefore we are unsure at the present and can not response with a definite answer.

c. Will there be a separate meta-analysis for each risk factor, and how will the authors determine when risk factors can be 'combined' e.g. would alcohol abuse/substance misuse/illicit drug use be considered one risk factor or several? I don't think it is necessary to list all the potential risk factors and how they will be managed, but it would be useful to include something about how this issue will be approached.

Again, results dependent, we aim to present risk factors grouped together by similar themes. This would follow a pattern much like that presented by Mars et al., (2019); demographic variables, psychosocial variables, psychiatric or mental health variables.

Mars, B., Heron, J., Klonsky, E. D., Moran, P., O'Connor, R. C., Tilling, K., ... & Gunnell, D. (2019). Predictors of future suicide attempt among adolescents with suicidal thoughts or non-suicidal self-harm: a population-based birth cohort study. *The Lancet Psychiatry*, 6(4), 327-337.

d. I am unclear what the authors mean by 'a sensitivity analysis will be run to assess the robustness of the studies'. Does this mean a sensitivity analysis where studies with lower quality ratings will be removed from the analysis?

We will exclude studies with low quality ratings from the analysis, yes.

Supplementary file 2 (search strategies)

15. The Embase search strategy does not appear to include all the terms listed on the previous page. In fact, it includes almost exclusively thesaurus terms and not keywords and does not include the second set of terms listed on the previous page (pertaining to risk factors/mechanisms etc). If possible, I would suggest that these terms not be included (as it appears they have not been in the Embase search) as their inclusion may limit the sensitivity of the search. And the inclusion of keywords as well as thesaurus terms is important to ensure sensitivity of the search.

Our search strategies were developed with an academic skills specialist, such that all key terms for Embase were explored and exploded to map those individual terms used in the Medline search. We have removed the Embase search to avoid confusion with potential readers and can be released on request to interested individuals.

VERSION 2 – REVIEW

REVIEWER	Sophie Epstein South London and Maudsley NHS Foundation Trust, UK
REVIEW RETURNED	16-Aug-2019

GENERAL COMMENTS	Many thanks for sending this protocol paper for a second review. The authors have addressed many of my comments, however there remain some areas which require further clarity. In some cases, the authors have provided responses to the comments but not included these clarifications in the manuscript its self so it would be helpful if these could be added to the manuscript to make it clearer for readers. 1. (point 8 in original review) Re the Newcastle Ottawa scale. The authors have stated that they will be using published versions. I appreciate this, but the scales still require adaptation for a specific study and parameters need to be decided on before conducting the quality assessment. Is the review from Bristol for which these versions have been used similar enough that identical versions could be used for this review? If not, some decisions will need to be made with regard to how the NOS will be used for this particular review and adapted versions included as a supplementary file.
---

	For example the 'comparability' item, asks for the most important confounders that the study should control for, and these need to be decided for the specific review being conducted. Re how the cohort is selected, it may be for example, that self-report is an appropriate measure for one review or outcome but may not be for another. Re 'follow-up long enough for outcomes to occur' – a decision needs to be made about how long is long enough. An adequate % follow up needs to be decided etc. These are just a few examples, but overall, there are quite a few decisions to be made with regard to how the NOS will be used for a particular study and these decisions should be made a priori. 2. (point 9 in original review) The authors have clarified the choice of age range and given justification – however I cannot see any of this included in the manuscript its self. Could at least a brief explanation be included in the paper? 3. (point 10 in original review) The authors have responded that not all participants will have had the same experience of self-harm, suicidal ideation or suicidal behaviour, but of course many participants will not have had any of these experiences – as this is the outcome being studied, some will experience the outcome and some will not. So stating that 'participants will also have had experiences of...' makes it sound as though ALL participants will experience the outcome of interest. 4. (point 11 in original review). The authors have explained that outcomes will be converted into binary variables, however this is not stated in the paper its self. In the paper, the author state 'A binary variable of self-harm, suicidal ideation and suicidal attempt will be taken as described by study authors' which suggests that the study authors will have already presented binary variables. This may not always be the case, so as stated in the author's response, it may be necessary to convert these to binary variables. Also the authors comment that they will include rating scales for suicidal behaviour/self-harm etc. but if these are continuous measures, it is not clear how (or if) these will be converted into binary variables. (if not, then the subsequent section which states that odds ratios will be the only effect measure used in meta-analysis will not be the case) If they will be converted, could authors please include something in the manuscript to explain how this will be done. 5. (Point 14 b in original review). I appreciate the authors may not yet know which studies will be possible to combine, but it would be good to include an acknowledgement in the manuscript that whether or not it is possible to pool studies will be contingent on them being similar enough in various ways. (point 14 c and d in original review). The authors have provided explanations here but I cannot see any of this included in the manuscript. Could the authors include a short explanation for how risk factors will be grouped, and also clarify what is meant by the sensitivity analysis within the manuscript. 6. (point 15 in original review). I understand that terms in the Embase search were developed to map onto the appropriate thesaurus, however using thesaurus terms in isolation risks missing important papers. The addition of keywords within the
--	---

	search strategies is also important. Secondly, rather than removing the Embase search, clarity is needed on whether or not the terms pertaining to risk factors/mechanisms were in fact included in the searches, as it appears they were not included in the Embase search. I am not clear what supplementary file 2 describes as it does not seem to be a list of the actual terms that were used in the searches (at least not the Embase search). 7. With regard to several of the points above, it may be helpful to address in the discussion section some of the challenges that are likely to be encountered, for example that it may be difficult to combine studies in meta-analysis if they do not examine similar enough exposures or outcomes, if it is not possible to convert outcomes to binary variables, if study designs are different etc. 8. I notice that in response to another reviewer's comment, the authors have included a statement that risk factors will be considered as predictors, moderators or mediators. This needs some clarity, as different studies will consider different factors as confounders (presumably this is what is meant by predictors?), moderators and mediators. How will each of these be handled in the review and meta-analysis? 9. Line 160 – I think 'was ran' should be 'was run' 10. Line 223 – 'if statistical pooling is not possible, a narrative synthesis will be conducted'. I imagine that in any case, not all of the studies identified will be possible to pool statistically. And each study may also contain additional data (such as additional risk factors, results reported as continuous outcomes etc) that it would be useful to describe in a narrative synthesis. The authors may wish to state that a narrative synthesis will be conducted in any case, plus meta-analyses of any groups of studies/results that are possible to pool.
--	---

VERSION 2 – AUTHOR RESPONSE

1. (point 8 in original review) Re the Newcastle Ottawa scale. The authors have stated that they will be using published versions. I appreciate this, but the scales still require adaptation for a specific study and parameters need to be decided on before conducting the quality assessment. Is the review from Bristol for which these versions have been used similar enough that identical versions could be used for this review? If not, some decisions will need to be made with regard to how the NOS will be used for this particular review and adapted versions included as a supplementary file.

For example the 'comparability' item, asks for the most important confounders that the study should control for, and these need to be decided for the specific review being conducted. Re how the cohort is selected, it may be for example, that self-report is an appropriate measure for one review or outcome but may not be for another. Re 'follow-up long enough for outcomes to occur' – a decision needs to be made about how long is long enough. An adequate % follow up needs to be decided etc. These are just a few examples, but overall, there are quite a few decisions to be made with regard to how the NOS will be used for a particular study and these decisions should be made a priori.

We have added the published NOS scales with their specific adjustments for this paper, as supplementary documents, per your suggestion.

Regarding the tool from Bristol, this was used for a systematic review regarding self-harm and suicide in Low- and Middle- Income Countries, therefore following the same topic. The tool condenses the NOS to 3 core areas. For the LMIC study, a high quality rating was offered if a study included consecutive series of suicidal behaviour cases where there were no threats to generalisability, and if the variable of interest was reached through a validated method; reliable measure or structured interview. The paper has recently been accepted and should be accessible shortly.

2. (point 9 in original review) The authors have clarified the choice of age range and given justification – however I cannot see any of this included in the manuscript its self. Could at least a brief explanation be included in the paper?

This has now been included.

Page 5.

3. (point 10 in original review) The authors have responded that not all participants will have had the same experience of self-harm, suicidal ideation or suicidal behaviour, but of course many participants will not have had any of these experiences – as this is the outcome being studied, some will experience the outcome and some will not. So stating that ‘participants will also have had experiences of...’ makes it sound as though ALL participants will experience the outcome of interest.

We would like to clarify that risk factors will only be extracted for suicidal or self-harm outcomes, and therefore all included participants will have these experiences. Risks for other outcomes such as depressive symptoms will not be extracted.

4. (point 11 in original review). The authors have explained that outcomes will be converted into binary variables, however this is not stated in the paper its self. In the paper, the author state ‘A binary variable of self-harm, suicidal ideation and suicidal attempt will be taken as described by study authors’ which suggests that the study authors will have already presented binary variables. This may not always be the case, so as stated in the author’s response, it may be necessary to convert these to binary variables. Also the authors comment that they will include rating scales for suicidal behaviour/self-harm etc. but if these are continuous measures, it is not clear how (or if) these will be converted into binary variables. (if not, then the subsequent section which states that odds ratios will be the only effect measure used in meta-analysis will not be the case) If they will be converted, could authors please include something in the manuscript to explain how this will be done.

Apologises for the misunderstanding surrounding this comment.

Self-harm and suicidal behaviour are inherently binary, as we are not considering severity. Suicidal ideation however, we will be using the originally used scale’s threshold as cut-off, if the population mean for ideation is above this threshold, ideation will be considered present, if the mean is below this threshold, ideation will be considered as absent.

Page 5.

5. (Point 14 b in original review). I appreciate the authors may not yet know which studies will be possible to combine, but it would be good to include an acknowledgement in the manuscript that

whether or not it is possible to pool studies will be contingent on them being similar enough in various ways.

(point 14 c and d in original review). The authors have provided explanations here but I cannot see any of this included in the manuscript. Could the authors include a short explanation for how risk factors will be grouped, and also clarify what is meant by the sensitivity analysis within the manuscript.

For the above two comments, we have now revised the manuscript to more clearly convey this information.

Page 6.

6. (point 15 in original review). I understand that terms in the Embase search were developed to map onto the appropriate thesaurus, however using thesaurus terms in isolation risks missing important papers. The addition of keywords within the search strategies is also important. Secondly, rather than removing the Embase search, clarity is needed on whether or not the terms pertaining to risk factors/mechanisms were in fact included in the searches, as it appears they were not included in the Embase search. I am not clear what supplementary file 2 describes as it does not seem to be a list of the actual terms that were used in the searches (at least not the Embase search).

Key words and terms were identified from previous systematic reviews regarding a) self-harm, b) suicide, c) LGBTQ+ populations or d) young people. These were collated into the original search; an EMBASE search was run which exploded all key words or terms. If this brought up new relevant words, these were added to the MEDLINE search as well. The search strategy was then reviewed with an academic skills librarian whom specialises in systematic review searches; this was to ensure that all keywords overlapped between search strategies and that all potential keywords were picked up. This was to ensure that the searches were as similar as possible. This was checked by all authors.

7. With regard to several of the points above, it may be helpful to address in the discussion section some of the challenges that are likely to be encountered, for example that it may be difficult to combine studies in meta-analysis if they do not examine similar enough exposures or outcomes, if it is not possible to convert outcomes to binary variables, if study designs are different etc.

We have included this point as a limitation for synthesis by meta-analysis.

Page 7.

8. I notice that in response to another reviewer's comment, the authors have included a statement that risk factors will be considered as predictors, moderators or mediators. This needs some clarity, as different studies will consider different factors as confounders (presumably this is what is meant by predictors?), moderators and mediators. How will each of these be handled in the review and meta-analysis?

In the case of a meta-analysis, risk factors will be considered by type of analysis in the original papers, such that those risks identified through mediation analysis will be clustered together and

likewise for other types of factors. If a risk is considered as a mediator in one study, and a moderator in another, it will be used in both analyses.

Page 6.

9. Line 160 – I think ‘was ran’ should be ‘was run’

We have now corrected this typo.

Page 5.

10. Line 223 – ‘if statistical pooling is not possible, a narrative synthesis will be conducted’. I imagine that in any case, not all of the studies identified will be possible to pool statistically. And each study may also contain additional data (such as additional risk factors, results reported as continuous outcomes etc) that it would be useful to describe in a narrative synthesis. The authors may wish to state that a narrative synthesis will be conducted in any case, plus meta-analyses of any groups of studies/results that are possible to pool.

This has now be changed within the manuscript to more readily suit the recommendation offered.

Page 7.